# Tuning of feedforward control enables stable muscle force-length dynamics after loss of autogenic proprioceptive feedback

Joanne C Gordon[1], Natalie C Holt[2], Andrew Biewener[3], Monica A Daley[1,4]*

[1]Comparative Biomedical Sciences, Royal Veterinary College, University of London, London, United Kingdom; [2]Evolution, Ecology & Organismal Biology, University of California, Riverside, Riverside, United States; [3]Organismic and Evolutionary Biology, Harvard University, Cambridge, Cambridge, United States; [4]Ecology and Evolutionary Biology, University of California, Irvine, Irvine, United States

**Abstract** Animals must integrate feedforward, feedback and intrinsic mechanical control mechanisms to maintain stable locomotion. Recent studies of guinea fowl (*Numida meleagris*) revealed that the distal leg muscles rapidly modulate force and work output to minimize perturbations in uneven terrain. Here we probe the role of reflexes in the rapid perturbation responses of muscle by studying the effects of proprioceptive loss. We induced bilateral loss of autogenic proprioception in the lateral gastrocnemius muscle (LG) using self-reinnervation. We compared in vivo muscle dynamics and ankle kinematics in birds with reinnervated and intact LG. Reinnervated and intact LG exhibit similar steady state mechanical function and similar work modulation in response to obstacle encounters. Reinnervated LG exhibits 23ms earlier steady-state activation, consistent with feedforward tuning of activation phase to compensate for lost proprioception. Modulation of activity duration is impaired in rLG, confirming the role of reflex feedback in regulating force duration in intact muscle.

*For correspondence: madaley@uci.edu

## Introduction

Sensory feedback is widely accepted as an integral component of vertebrate locomotor control (*Cohen, 1992*; *Donelan and Pearson, 2004*; *Grillner, 2011*; *Prochazka and Ellaway, 2012*; *Rossignol et al., 2006*). Proprioception from muscle mechanoreceptors contributes to 1) short-latency reflexes via spinal mono- and polysynaptic pathways to regulate the ongoing activity and muscle mechanical output (force, stiffness, impedance and work) and 2) longer-latency responses to coordinate and maintain task-level goals for balance and movement (*Grillner, 2011*; *Lam and Pearson, 2002*; *Frigon and Rossignol, 2006*; *Prochazka and Ellaway, 2012*; *Proske and Gandevia, 2012*; *Rossignol et al., 2006*; *Safavynia and Ting, 2013*; *Sherrington and Laslett, 1903*; *Sherrington, 1910*). Proprioceptive reflexes can occur through *autogenic* (self-generated) pathways arising from the same muscle, and through *heterogenic* pathways arising from other muscles via spinal interneurons (*Abelew et al., 2000*; *Frigon and Rossignol, 2006*; *Lam and Pearson, 2002*; *Nichols, 1989*; *Ross and Nichols, 2009*). Thus, the relationship between a specific sensory signal and its resulting effects is complex and dynamic.

Despite recognized functions of proprioception, the relative contribution of feedback control in high-speed locomotion remains unclear. Sensorimotor delay constrains how quickly an animal can sense and respond to a stimulus using feedback control (*More and Donelan, 2018*; *More et al., 2010*). The fastest possible feedback loop occurs through mono-synaptic reflexes, which involve a delay that increases in proportion to nerve transmission distance. This reflex delay becomes a larger

fraction of the stride cycle with increasing speed, limiting time available for reflex-mediated correction.

The challenges of long delays relative to stride cycle times likely necessitates greater reliance on feedforward control strategies at higher speeds. Here we use feedforward to refer to the contributions to motor output arising from the motor cortex, descending pathways and rhythmic spinal networks (*Frigon and Rossignol, 2006*; *Pearson, 2000*; *Yakovenko et al., 2004*). Rhythmic spinal networks can generate the basic flexion and extension motor pattern for gait, even when proprioceptive feedback is removed (e.g., *Pearson et al., 2003*; *Sharp and Bekoff, 2015*). Normally, however, descending networks act in concert with spinal networks and feedback, using multimodal and distributed sensory inputs to update state estimates, regulate rhythm and control foot placement (*Cohen, 1992*; *Drew and Marigold, 2015*; *Marigold and Drew, 2017*; *Pearson, 2000*; *Roth et al., 2014*; *Todorov, 2004*; *Potocanac et al., 2014*; *Wolpert et al., 2011*). Consequently, there is no 'pure' feedfoward control within vertebrate systems. We use the term here as a pragmatic distinction, where *feedforward* refers to anticipatory 'look-ahead' control over one or more stride cycles, and *feedback* refers to reflex-mediated reactive responses to perturbations.

Although feedforward networks normally act in concert with feedback, feedforward motor activation coupled to intrinsic muscle properties can be sufficient to produce stable gait (*Yakovenko et al., 2004*). Consistent with this, the lateral gastrocnemius (LG) of guinea fowl (*Numida meleagris*) rapidly absorbs energy in response to unexpected drop perturbations (*Daley et al., 2009*), stabilizing high speed running without a reflex response. The rapid perturbation response arises from the intrinsic mechanical properties of the muscle-tendon tissues and musculoskeletal system (*Brown and Loeb, 2000*; *Loeb et al., 1999*; *Jindrich and Full, 2002*; *Azizi et al., 2008*). Intrinsic mechanical responses can be actively tuned by the specific feedforward pattern of muscle activation. For example, humans hopping on surfaces with randomized, sudden increases in ground stiffness show a feedforward increase in muscle co-activation and knee flexion, increasing mechanical stability (*Moritz and Farley, 2004*). However, many perturbation responses involve multiple control mechanisms that overlap in time. Guinea fowl running over obstacles use a combination of feedforward, intrinsic mechanical and reflex-mediated mechanisms, with a delay of ~40 ms for reflex-mediated increases in muscle force (6ms reflex latency + 34ms force development delay: *Daley et al., 2009*; *Daley and Biewener, 2011*). Considering that the feedforward and intrinsic mechanical contributions alter ongoing muscle dynamics *before* the reflex-mediated response, it is difficult to disentangle the specific contributions of proprioceptive reflexes to the observed perturbation responses (*Daley and Biewener, 2011*; *Gordon et al., 2015*).

## Investigating the role of proprioception through self-reinnervation

Here we probe the integration of feedforward, feedback and intrinsic mechanical control by eliciting a proprioceptive deficit in the lateral gastrocnemius muscle (LG) of the guinea fowl (*Numida meleagris*) using bilateral self-reinnervation (*Figure 1*). Self-reinnervation involves peripheral nerve branch transection and immediate repair, resulting in recovery of motor output with long-term, local loss of autogenic muscle proprioception (*Cope et al., 1994*; *Bullinger et al., 2011*). Self-reinnervation occurs through axonal regrowth and reconnection with denervated tissues over a recovery period of 4-8 weeks (*Carr et al., 2010*; *Cope et al., 1994*; *Gordon and Stein, 1982*; *Vannucci et al., 2019*). Reinnervated muscles retain a deficit in the monosynaptic stretch reflex due to synaptic retraction of primary muscle spindle afferents and disconnection from parent motoneuron populations (*Alvarez et al., 2011*; *Brandt et al., 2015*). However, intermuscular force and length feedback networks may remain partially intact (*Lyle et al., 2016*). Cats and rats with reinnervated muscles maintain whole-limb function by adjusting inter-joint coordination and muscle co-activation to compensate for loss of reflex-mediated ankle stiffness (*Abelew et al., 2000*; *Maas et al., 2007*; *Chang et al., 2009*; *Boeltz et al., 2013*). These findings highlight the ability of animals to flexibly exploit musculoskeletal plasticity to maintain function and suggest self-reinnervation as a promising tool to investigate sensorimotor control mechanisms.

Studying neuromuscular control in the guinea fowl, a bipedal animal model, provides insight into similarities and differences among vertebrates that may relate to locomotor modality, evolutionary history, or both. Birds share features of sensorimotor structure and function with mammals, including muscle tissue properties (*Nelson et al., 2004*; *Poore et al., 1997*) and muscle proprioception through muscle spindle and Golgi tendon organs (*Dorward, 1970*; *Haiden and Awad, 1981*;

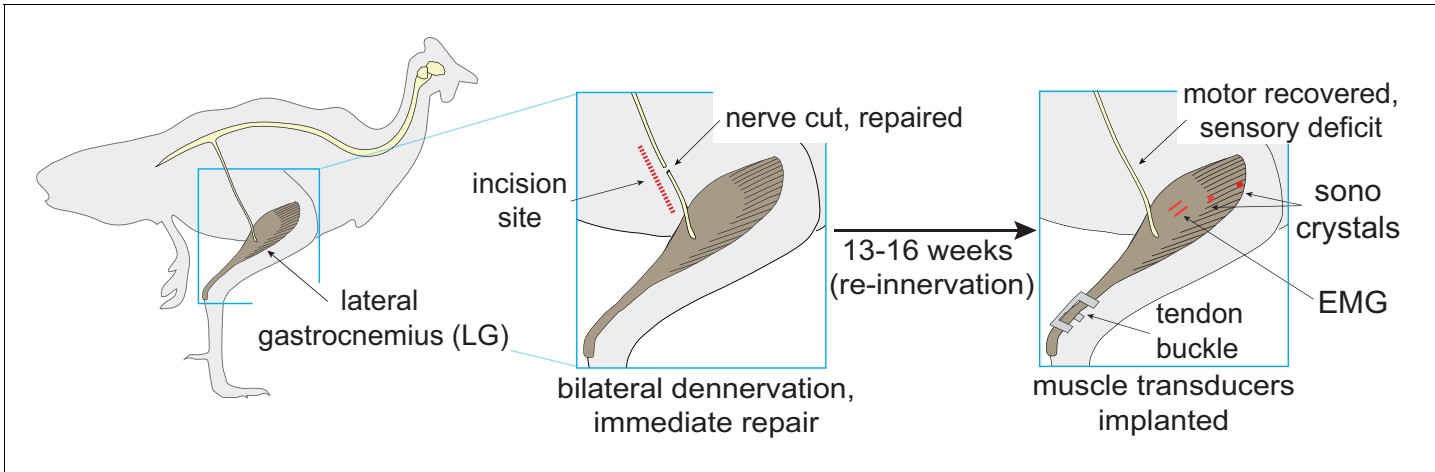

**Figure 1.** Reinnervation protocol. Procedure for bilateral self-reinnervation of the lateral gastrocnemius (LG), followed by transducer implantation for in vivo recordings of muscle force (tendon buckle), fascicle length (sonomicrometry crystals) and electromyographic activity (EMG).

*Maier, 1992*). Ground birds use bipedal walking and running gaits with mechanics and energetics similar to human locomotion (*Heglund et al., 1982*; *Taylor et al., 1982*; *Gatesy and Biewener, 1991*; *Roberts et al., 1997*; *Daley and Birn-Jeffery, 2018*). Bipedal gaits involve substantial periods of single-limb contact, which limits the redundancy of balance mechanisms and poses a challenge for stability (*Daley et al., 2009*; *Clark and Higham, 2011*; *Daley and Biewener, 2011*). Whereas a quadrupedal cat or rat might be able to compensate for deficits by shifting weight bearing among legs, a biped with a bilateral proprioceptive deficit cannot. Accordingly, one goal of the current study is to explore whether or not guinea fowl exhibit a similar response to proprioceptive deficit as observed in quadrupedal vertebrates (*Abelew et al., 2000*; *Maas et al., 2007*; *Chang et al., 2009*; *Boeltz et al., 2013*).

We hypothesize that autogenic proprioceptive deficit will lead to increased reliance on feedforward control mechanisms and intrinsic muscle mechanics to maintain stable locomotion. To test for shifts in stability and control mechanisms, we measured ankle kinematics and in vivo LG muscle dynamics (length, force and activation) during treadmill running on level and obstacle terrain. There are several potential mechanisms to compensate for autogenic proprioceptive deficit: 1) Birds might compensate for proprioceptive deficit by increasing feedforward muscle activation before obstacle contact, as observed in birds negotiating high-contrast obstacles (*Gordon et al., 2015*). 2) Alternatively, if feedback regulation of LG is essential for stability in fast locomotion, loss of autogenic proprioception may necessitate increased reliance on heterogenic reflex pathways from synergists, with a slight delay compared to intact animals, as suggested by work in cats and rats (*Boeltz et al., 2013*; *Lyle et al., 2016*). 3) Finally, if intrinsic mechanics are mainly responsible for the modulation of muscle force and work, we might expect minimal change in muscle activity patterns (EMG), as observed in birds subjected to unexpected drop perturbations (*Daley et al., 2009*). We expect birds to compensate for proprioceptive deficit by tuning gait and feedforward muscle activity to maintain a stable response to obstacle perturbations, as observed in reinnervated rats and cats walking on slopes (*Abelew et al., 2000*; *Maas et al., 2007*; *Chang et al., 2009*; *Boeltz et al., 2013*). If stability is impaired following reinnervation, this should be evident from increased variance and longer time to recover from obstacles. By investigating the shifts in guinea fowl LG muscle force, length and activation dynamics following reinnervation, we hope to gain insight into the mechanisms of sensorimotor integration and plasticity that enable robustly stable and agile bipedal locomotion.

## Results

### Mechanical function of intact *versus* reinnervated LG

We find that many features of the steady-state in vivo mechanical function of the guinea fowl reinnervated lateral gastrocnemius (rLG) are similar to that previously measured in the intact LG

**Table 1.** Statistical results.

F-statistics for linear mixed effect model ANOVA with fixed effects of treatment cohort (*treatment*: intact, reinnervated) and stride category (*stride ID*) and the interaction *treatment* x *stride ID* on measures of muscle contraction mechanics and activation. Bolding indicates statistical significance using FDR corrected threshold (p<=0.0263, see Methods). Degrees of freedom for fixed effects were *treatment* = 1, *stride ID* = 4, *interaction* = 4, and error = 2529. See *Table 1—source data 1* for p-values.

| | F-statistic | | |
| Variable | *treatment* | *stride ID:* | *interaction* |
| --- | --- | --- | --- |
| $W_{net}$ | 2.44 | 172.04 | 7.44 |
| $F_{pk}$ | 0.52 | 30.12 | 61.73 |
| $L_{pkF}$ | 1.52 | 238.03 | 39.52 |
| $V_{pkF}$ | **11.43** | 28.09 | 8.73 |
| $T_{force}$ | 0.27 | 99.09 | 18.32 |
| $T_{stride}$ | 0.10 | 12.17 | 17.82 |
| $E_{tot}$ | 0.01 | 49.05 | 3.51 |
| $E_{freq}$ | 3.93 | 8.71 | 1.85 |
| $E_{phase}$ | **5.72** | 2.34 | 7.64 |
| $E_{dur}$ | **10.02** | 8.86 | 10.69 |

The online version of this article includes the following source data for Table 1:

**Source data 1.** P-values linear mixed effect model ANOVA with fixed effects of treatment cohort (*treatment*: intact, reinnervated) and stride category (*stride ID*) and the interaction *treatment* x *stride ID*.

(iLG) (*Table 1*). In *Figure 2*, average trajectories (mean±95% confidence interval) are shown for muscle strain, force and electromyographic activity (iLG at top, rLG below), with the average for steady level running in grey. During the swing phase of the stride cycle, both iLG and rLG exhibit a period of passive stretch, followed by rapid shortening. Activation and force development begin in late swing around the time of the transition from stretch to shortening, initiating rapid active shortening until foot contact (*Figure 2*, triangles). At the time of foot contact, force increases rapidly to a peak before midstance, then declines more slowly. Typically, in level running both iLG and rLG show a near-isometric phase in early stance, followed by shortening in late stance, which produces net positive work, as indicated by counter-clockwise force-length work loops (*Figure 3*). The average magnitude of work output ($W_{net}$) during steady level running is similar between the iLG and rLG, with similar spread of the distribution around the mean (*Figure 4*). However, rLG shows faster shortening velocity at peak force ($V_{pkF}$) compared to the iLG across both level and obstacle terrains (*Figure 4*), indicating a difference in steady state contraction dynamics.

## Force-length dynamics and work output during obstacle negotiation

In obstacle encounters (*Figure 2*, S 0), foot contact with the obstacle occurs earlier in the stride cycle compared to level terrain, altering force-length dynamics during the obstacle stance period. During obstacle contact, both iLG and rLG remain at longer lengths, force increases rapidly to reach a higher peak force, and the muscle shortens throughout force development, producing positive work (*Figure 2*, S 0). Both iLG and rLG exhibit increased force and work output in obstacle strides compared to level strides (*Figure 3*, S 0). The magnitude of the shift in work output ($W_{net}$) in obstacle strides (S 0) is similar between intact and reinnervated cohorts, increasing by 3.60±0.57 $Jkg^{-1}$ in iLG and 3.88±0.60 $Jkg^{-1}$ in rLG (mean±95% ci, *Figure 4B*, *Table 2* ).

Although the magnitude of the shifts in work output are similar between iLG and rLG, the mechanisms underlying the shift in work output differ between them (*Figure 4)*. In iLG, increased work upon obstacle contact occurs through modest increases in both peak force ($F_{pk}$) and shortening velocity ($V_{pkF}$), compared to level strides. In contrast, rLG exhibits a substantially larger increase in $F_{pk}$ on obstacle strides and maintains similar $V_{pkF}$ between level and obstacle terrain strides (S 0, *Figure 4B*, *Table 2*). Reinnervated LG also exhibits small but significant increases in $F_{pk}$ in the strides preceding and following obstacle contact (S-1 and S+1, respectively), compared to level terrain.

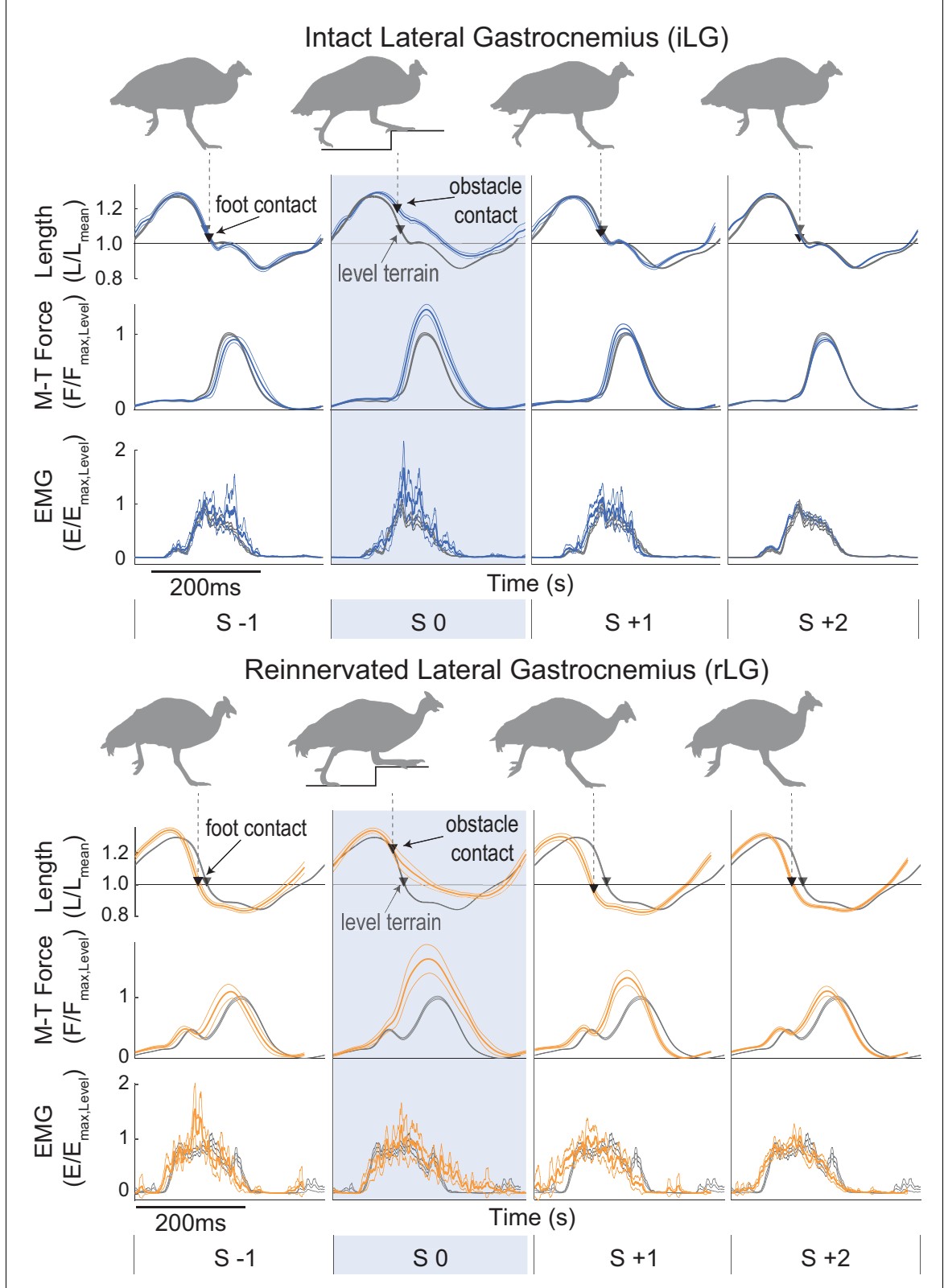

**Figure 2.** Muscle trajectories during obstacle negotiation for intact and reinnervated lateral gastrocnemius (iLG: blue, top, rLG: orange, bottom). Stride cycle averages are shown, from mid-swing to mid-swing (mean ± 95% ci) for a 4-stride sequence in obstacle terrain, with steady level terrain means as a reference, in grey. The shaded box is an obstacle encounter (S 0). Obstacle terrain strides are coded as in *Daley and Biewener, 2011*, for strides preceding (S −1), on (S 0) and following obstacle contact (S +1), with S +2 including all other strides between obstacles. Trajectories are fractional

*Figure 2 continued on next page*

*Figure 2 continued*

muscle fascicle length (top), muscle-tendon force (middle) and rectified myoelectric activity (EMG). Triangles indicate the timing of foot-ground contact (grey: level terrain, black: obstacle terrain). Example data is shown from one individual in each treatment cohort. See *Figure 2—figure supplement 1* for details on stride-cycle cutting and categorization in an example stride sequence in obstacle terrain.

The online version of this article includes the following figure supplement(s) for figure 2:

**Figure supplement 1.** Example 6-stride sequence of in vivo muscle recordings of the reinnervated lateral gastrocnemius (rLG) in the right leg, running at 1.7 ms$^{-1}$ on the obstacle treadmill.

## Shifts in activation patterns between intact and reinnervated LG

Despite deficits in LG monosynaptic reflex following reinnervation, rLG and iLG show similar increases in total muscle activation intensity ($E_{tot}$, integral of EMG) in obstacle strides compared to steady level strides (S 0, *Figure 5*). Intact LG exhibits a 4% increase in duration in obstacle strides (S 0) compared to level; however, there is no significant increase in EMG duration for rLG in S 0 (*Figure 5B*, *Table 2*). This suggest that the observed increase in $E_{tot}$ in rLG obstacle strides occurs through increased activation amplitude, not increased duration.

Several results suggest a shift in central drive and feedforward activation pattern in rLG compared to iLG. Reinnervated LG exhibits longer steady-state duration of activity ($E_{dur}$) compared to iLG across all level and obstacle terrain strides, averaging 37% of the stride cycle in rLG compared to 29% in iLG (*Figure 5B*, *Table 2*). Additionally, rLG exhibits higher average frequency of EMG activity across all strides compared to the intact cohort (*Figure 5*, *Table 2*, *Figure 5—figure supplement 1*). Finally, the steady-state timing of rLG activation is phase-shifted to 6% (23ms) earlier in the stride cycle relative to the length trajectory, quantified by the variable '$E_{phase}$' (*Figure 6*, *Tables 1–*

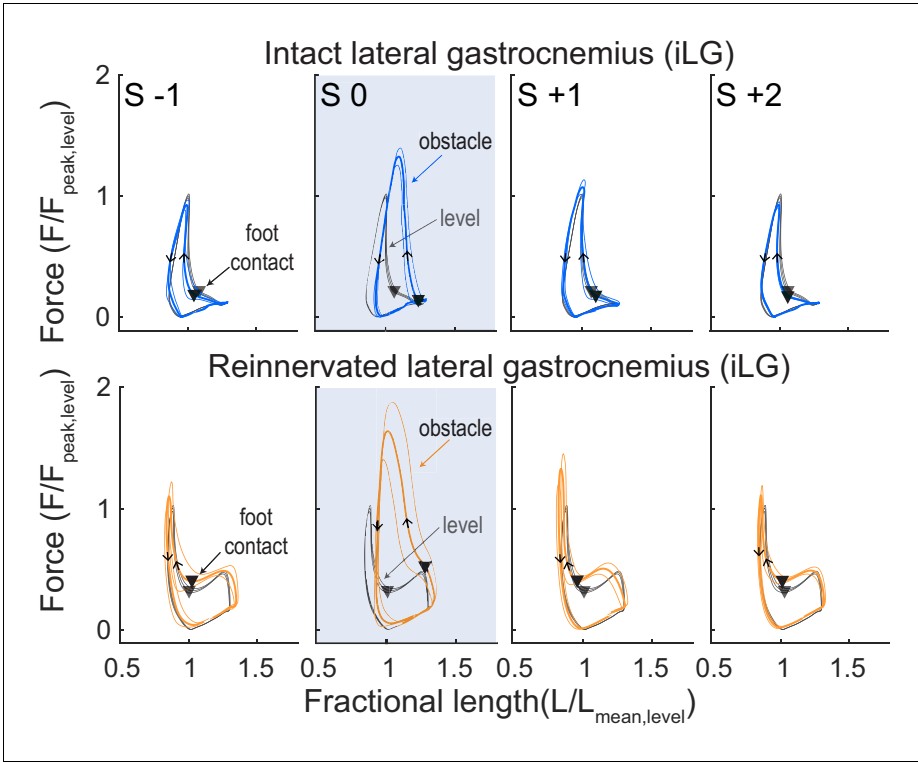

**Figure 3.** Force-length work loops for iLG (top) and rLG (bottom), for a single individual from each treatment cohort (intact/reinnervated, mean ±95% ci). Level mean in grey and obstacle strides in colored lines (iLG: blue, rLG: orange). Stride categories as in *Figure 2*, where the shaded box is an obstacle encounter (S 0). Triangles indicate the timing of foot-ground contact and arrows indicate the direction of the work loop, with a counter-clockwise loop corresponding to net positive muscle work.

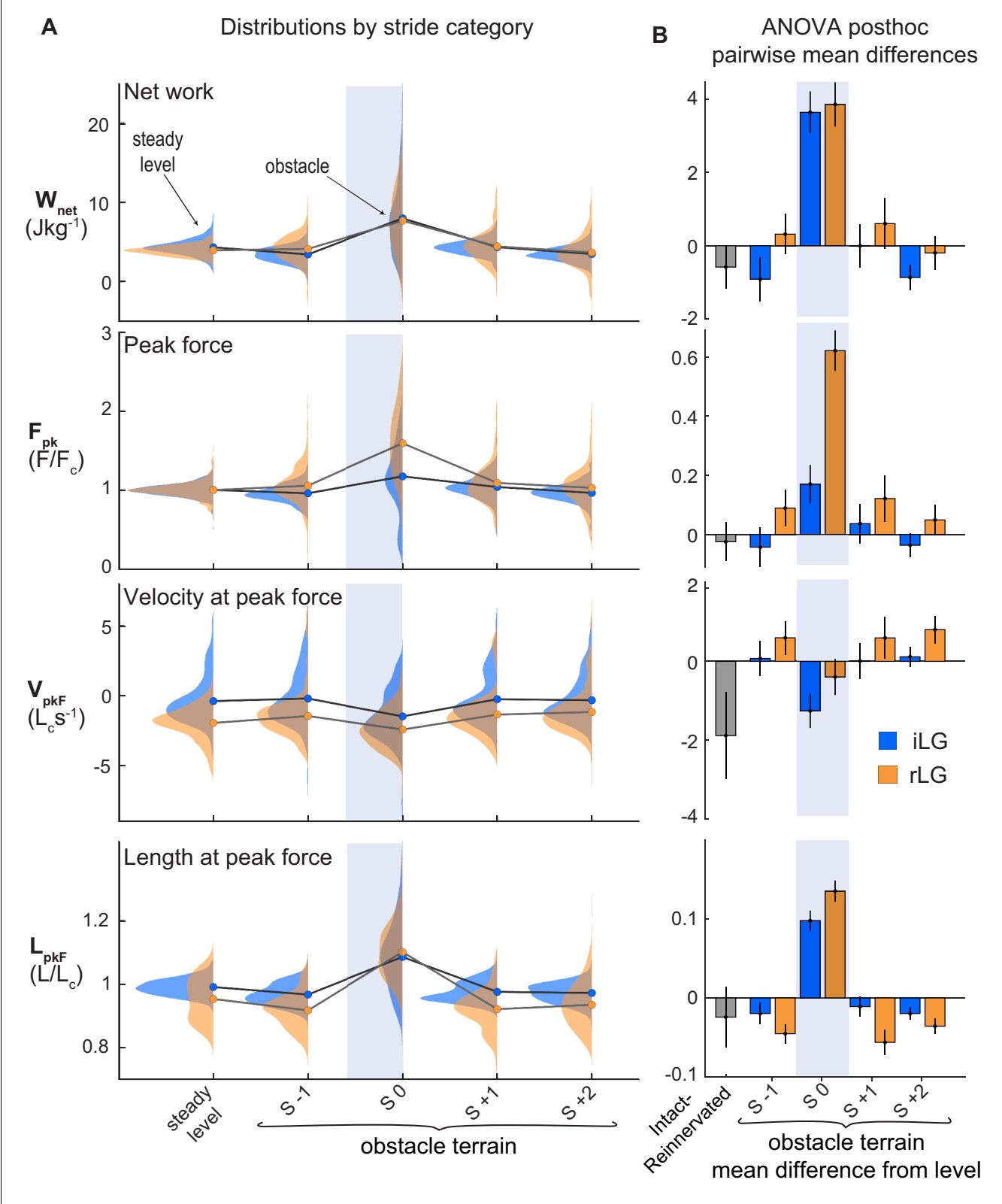

**Figure 4.** LG muscle mechanical output during obstacle negotiation. (A) Distributions of muscle total work output ($W_{net}$), peak force ($F_{pk}$), velocity and length at peak force ($V_{pkF}$, $L_{pkF}$) across stride categories for iLG (blue) and rLG (orange). Circles indicate group means. Lines connect means between stride categories, to highlight the shifts in relation to obstacle encounters (S 0). (B) Pairwise mean differences (mean ±95% ci) for fixed effect categories,

*Figure 4 continued on next page*

*Figure 4 continued*

between intact and reinnervated treatment cohorts (grey bar), and between obstacle stride categories compared to level means, within treatment cohorts (colored bars). See *Tables 1* and *2* for full statistics results and summary data.

*2*). Earlier activation onset may help explain the higher rate of shortening in rLG compared to iLG, reported above.

## Timing of obstacle-induced changes in EMG activity in iLG and rLG

To explore the timing of obstacle-induced shifts in EMG activity relative to perturbations in force and length, we calculated a difference trajectory between the steady-state level and obstacle perturbed stride cycles (S 0 – Lev) for each individual, and then calculated the mean and 95% confidence interval across individuals (*Figure 7*). Increased EMG activity begins ~30-40 ms *before* obstacle-induced increases in length and force, for both iLG and rLG, suggesting an anticipatory (feedforward) contribution to increases in $E_{tot}$ (*Figure 7*, arrows indicating 'anticipatory increase'). In iLG, the anticipatory increase in EMG starts ~25% of stride period (asterisk in EMG trace and vertical dashed line in *Figure 7A*). Starting around 58% of stride period, there is another distinct burst of increased activity, suggesting reflex-mediated contribution to increased EMG in obstacle strides (*Figure 7*, arrow indicating 'reflex'). In rLG, the 'anticipatory increase' in EMG starts around 21% of stride (asterisk and vertical dashed line in 7B); however activity in the latter half of stance is highly variable and idiosyncratic among individuals (*Figure 7—figure supplement 1*), as indicated by the wide 95% confidence intervals spanning the region of time where the iLG shows a distinct reflex response (compare *Figure 7B* versus 7A lower panels). Cross-correlation between the obstacle perturbation trajectories for iLG reveals a correlation of 0.82 between length and EMG deviations, and a correlation of 0.85 between force and EMG deviations. For rLG, the cross-correlations are reduced to 0.53 between length and EMG deviations, and 0.57 between force and EMG deviations, respectively. The reduced correlations suggest a disrupted reflex-mediated response to muscle load and strain in the latter half of stance in rLG, but which is present in iLG (*Figure 7*). Considering that the changes in muscle length and force are strongly correlated with each other during obstacle encounters in both intact and reinnervated conditions (*Figure 7*), it is difficult to distinguish the specific sensory signal eliciting reflex responses.

**Table 2.** Pairwise posthoc comparisons.

Pairwise mean differences (mean ± 95% ci) between intact and reinnervated treatment cohorts (left), and between obstacle stride categories compared to level stride means, within treatment cohorts (intact/reinnervated). Bolding indicates statistical significance using FDR corrected threshold (p <= 0.0263). See *Table 2—source data 1* for p-values.

| Variable | Treatment cohort | Intact | | | | Reinnervated | | | |
|---|---|---|---|---|---|---|---|---|---|
| | | S −1 | S 0 | Str +1 | S +2 | S −1 | S 0 | Str +1 | S +2 |
| $W_{net}$ | −0.47 ± 0.59 | **−0.91 ± 0.60** | **3.60 ± 0.57** | 0.00 ± 0.59 | **−0.86 ± 0.34** | 0.34 ± 0.55 | **3.88 ± 0.60** | 0.49 ± 0.70 | −0.19 ± 0.46 |
| $F_{pk}$ | −0.02 ± 0.06 | −0.04 ± 0.07 | **0.17 ± 0.06** | 0.04 ± 0.07 | −0.04 ± 0.04 | **0.09 ± 0.06** | **0.62 ± 0.07** | **0.10 ± 0.08** | 0.05 ± 0.05 |
| $L_{pkF}$ | −0.02 ± 0.04 | −0.02 ± 0.01 | **0.10 ± 0.01** | −0.01 ± 0.01 | **−0.02 ± 0.01** | **−0.05 ± 0.01** | **0.14 ± 0.01** | **−0.06 ± 0.02** | **−0.04 ± 0.01** |
| $V_{pkF}$ | −1.89 ± 1.10 | 0.07 ± 0.45 | **−1.26 ± 0.43** | 0.00 ± 0.45 | 0.11 ± 0.26 | **0.60 ± 0.42** | −0.39 ± 0.46 | **0.63 ± 0.53** | **0.81 ± 0.35** |
| $T_{force}$ | 0.02 ± 0.07 | −0.01 ± 0.02 | **0.11 ± 0.02** | −0.01 ± 0.02 | −0.01 ± 0.01 | **−0.07 ± 0.02** | **0.07 ± 0.02** | **−0.06 ± 0.02** | **−0.06 ± 0.02** |
| $T_{stride}$ | 0.00 ± 0.02 | −0.01 ± 0.02 | **0.03 ± 0.02** | **−0.02 ± 0.02** | 0.01 ± 0.01 | **−0.07 ± 0.02** | **0.04 ± 0.02** | **−0.04 ± 0.03** | **−0.03 ± 0.02** |
| $E_{tot}$ | 0.01 ± 0.22 | 0.12 ± 0.18 | **0.76 ± 0.18** | 0.11 ± 0.18 | **0.12 ± 0.11** | 0.16 ± 0.17 | **0.74 ± 0.19** | **0.39 ± 0.22** | 0.09 ± 0.14 |
| $E_{freq}$ | 55.70 ± 55.12 | 3.05 ± 9.44 | **−15.05 ± 9.04** | −1.46 ± 9.34 | −0.46 ± 5.43 | −2.86 ± 8.75 | **−22.06 ± 9.52** | −6.81 ± 11.04 | **−7.68 ± 7.28** |
| $E_{phase}$ | **−0.06 ± 0.05** | −0.01 ± 0.02 | 0.00 ± 0.02 | 0.00 ± 0.02 | −0.01 ± 0.01 | 0.01 ± 0.02 | **0.04 ± 0.02** | −0.01 ± 0.02 | 0.00 ± 0.01 |
| $E_{dur}$ | **0.08 ± 0.05** | 0.00 ± 0.03 | **0.04 ± 0.02** | −0.01 ± 0.02 | 0.00 ± 0.01 | **−0.04 ± 0.02** | −0.01 ± 0.03 | 0.00 ± 0.03 | **−0.03 ± 0.02** |

The online version of this article includes the following source data for Table 2:

Source data 1. P-values for posthoc pairwise mean differences between intact and reinnervated treatment cohorts (left column) and between obstacle stride categories compared to the level terrain means, within treatment cohorts (intact/reinnervated).

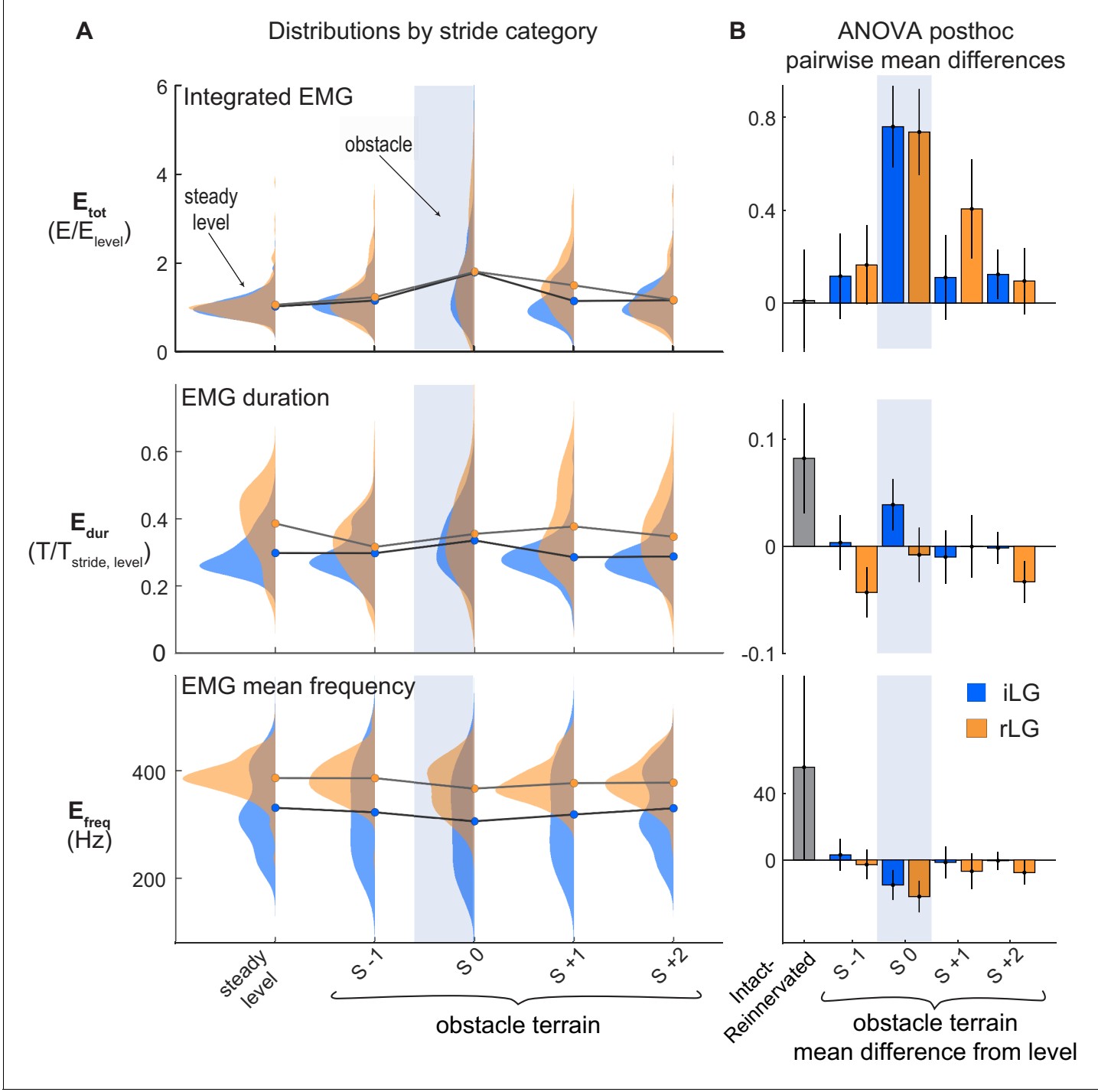

**Figure 5.** LG muscle activation during obstacle negotiation. (**A**) Distributions of total intensity EMG activity ($E_{tot}$), duration of activity ($E_{dur}$) and mean frequency of activity ($E_{freq}$) across stride categories for iLG (blue) and rLG (orange). Circles indicate group means. Lines connect means between stride categories, to highlight the shifts in relation to obstacle encounters (S 0). (**B**) Pairwise mean differences (mean ±95% ci) for fixed effects, as presented as in *Figure 4*. See *Tables 1* and *2* for full statistics results and summary data.

The online version of this article includes the following figure supplement(s) for figure 5:

**Figure supplement 1.** LG frequency distribution.

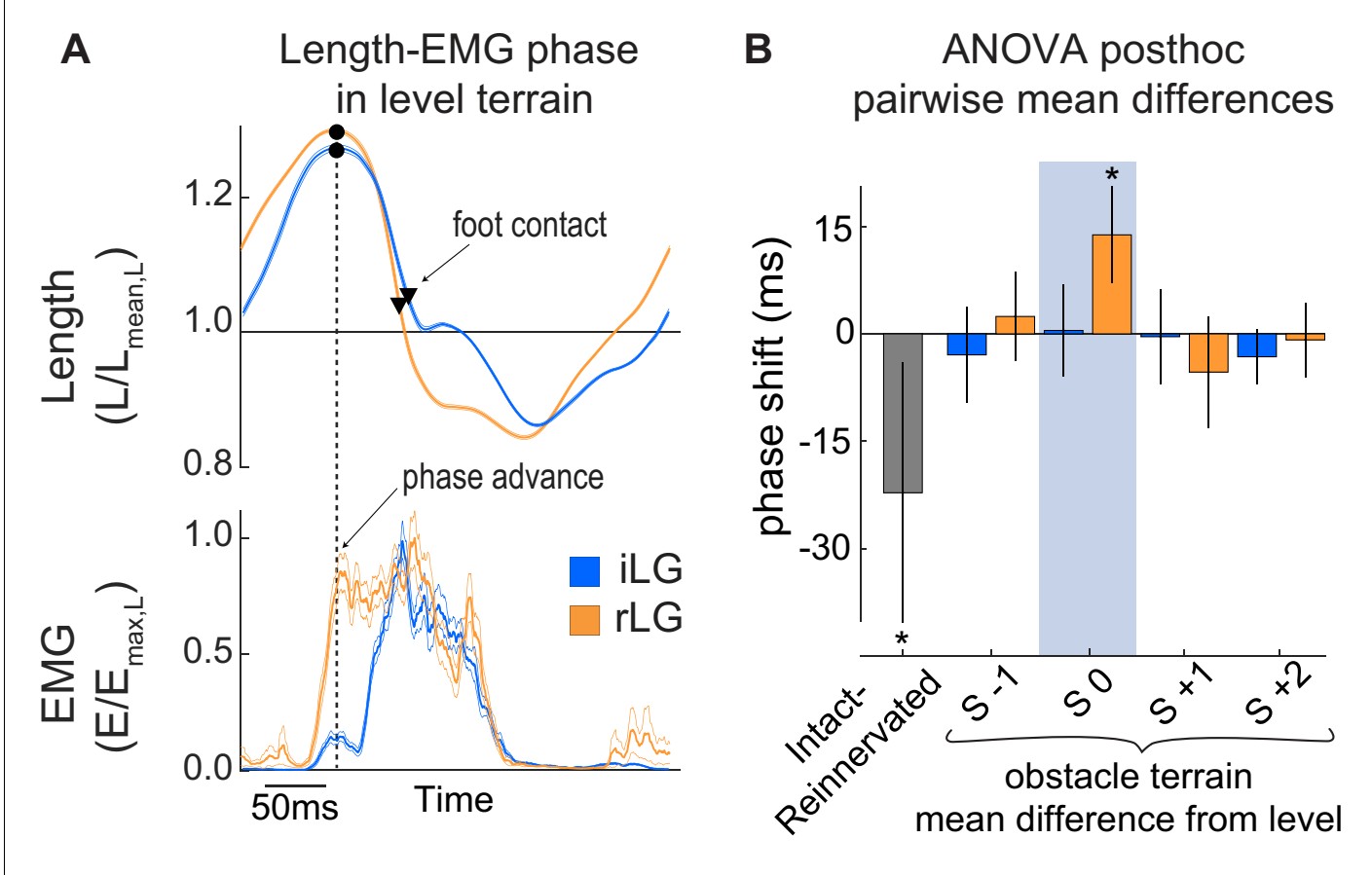

**Figure 6.** Phase relationship ($E_{phase}$) between length and EMG activation. (**A**) Average steady-state length and activation trajectories for iLG and rLG in level terrain, aligned in time based on peak length during the swing phase, before foot-substrate contact. Black dot and vertical dashed line indicate the time of peak fascicle length. Triangles indicate timing of foot contact. (**B**) Pairwise mean differences in $E_{phase}$ (mean ±95% ci) between intact and reinnervated treatment cohorts (grey), and obstacle stride categories compared to level terrain within each cohort (colored bars). $E_{phase}$ is reported in the ANOVA tables as a fraction of the stride cycle but is reported in milliseconds here.

### Stability and kinematic changes during obstacle negotiation in intact vs reinnervated birds

Compared to the intact cohort, birds with rLG exhibit more pronounced shifts in gait dynamics in obstacle terrain relative to level terrain. Obstacle-induced increases in peak force ($F_{pk}$) are larger for rLG compared to iLG (S 0 *Figure 4*, *Table 2*), reflecting larger deviations from steady state in response to the same obstacle. Additionally, rLG shows small but significant increases in peak force ($F_{pk}$) in the strides preceding and following obstacle contact (S -1, S +1) compared to steady level strides (*Figure 4B*, *Table 2*). Multiple significant differences from level strides occur for rLG in S +1, including a 39±22% increase in $E_{tot}$, a 6±2% decrease in force duration and a 4±3% decrease in stride duration (mean±95% ci, *Table 2*). In contrast, most variables for iLG have recovered to steady state in S +1 (*Table 2*). Both iLG and rLG rapidly increase work output during obstacle encounters and face increased activation costs for locomotion in obstacle terrain. However, rLG shows larger deviations from steady state, and a slower recovery to steady state mechanics and activation level compared to iLG, indicating reduced stability.

Reinnervated birds also show differences in running kinematics in obstacle terrain compared to the intact cohort, undergoing a more pronounced increase in ankle flexion in obstacle encounters (*Figure 8*). Reinnervated birds also use a shorter stride duration immediately preceding the obstacle encounter (S -1), suggesting anticipatory preparation that is not observed in intact birds (*Figure 8*). These observations suggest reduced ankle stiffness and increased anticipatory preparation for obstacle encounters in the reinnervated birds.

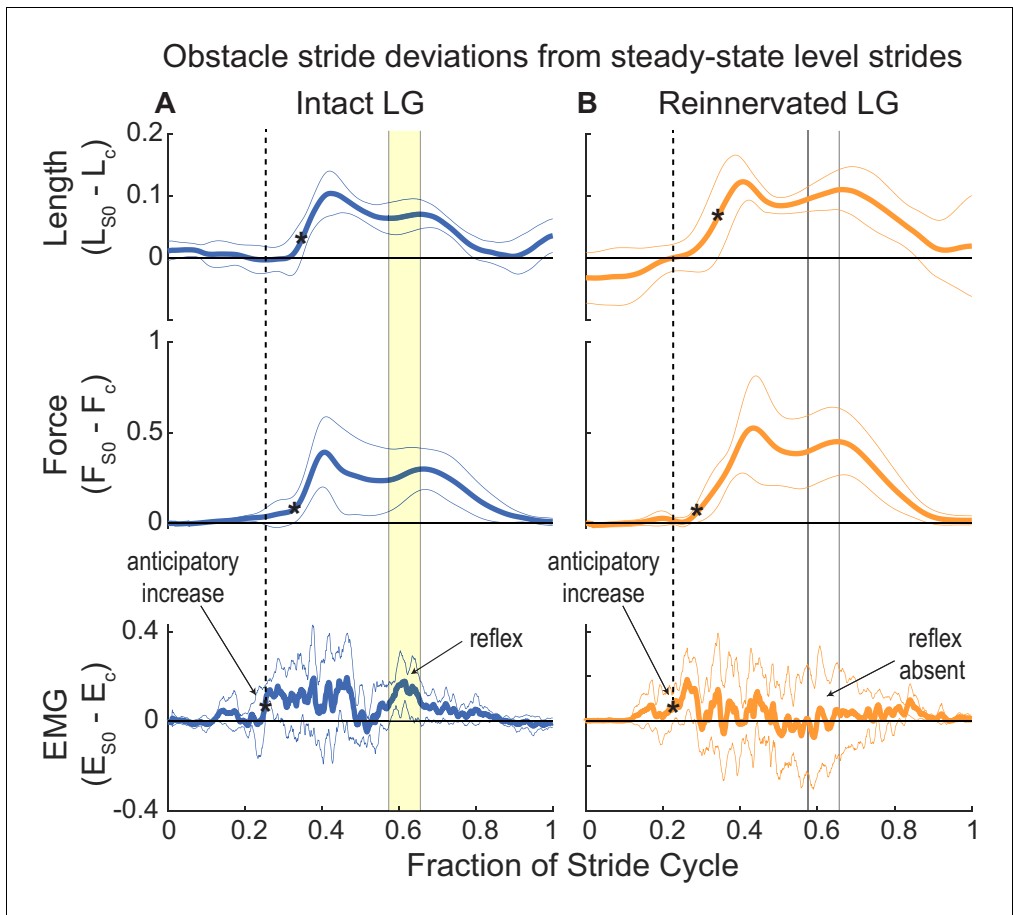

**Figure 7.** Deviations from steady state in the stride cycle trajectories of muscle length, force and activation, between obstacle strides (S 0) and level strides (grand mean ±95% ci across individuals). The horizontal zero line indicates no difference from steady state in S 0. The stride cycle is from mid-swing to mid-swing, as in *Figure 2*. A black asterisk (*) indicates the first timepoint in each trajectory that differs significantly from the level mean. The dashed vertical line and arrow indicating 'anticipatory increase' highlights a significant increase in EMG that starts *before* deviations length and force in S 0. In (**A**) (iLG), solid vertical lines and yellow fill indicates a 2nd period of significantly increased EMG in late stance that correlates with increased fascicle length and force, suggesting a reflex response. In (**B**) (rLG), the anticipatory increase in EMG is present; however, wide confidence intervals for EMG in late stance indicates inconsistent patterns of activity across individuals, despite similar increases in length and force as iLG. This suggests disrupted autogenic feedback and idiosyncratic heterogenic feedback patterns across individuals (*Figure 7—figure supplement 1*).

The online version of this article includes the following figure supplement(s) for figure 7:

**Figure supplement 1.** Muscle trajectories during obstacle negotiation for all individuals in the reinnervated cohort.

## Discussion

### What is the role of proprioception in the control of high-speed locomotion?

We investigated the role of reflexes in the sensorimotor control of running by examining the effects of proprioceptive deficit on the mechanical function of the lateral gastrocnemius muscle (LG) of guinea fowl. Long sensorimotor delays relative to limb cycling times necessitate that animals use a combination of feedforward, feedback and intrinsic mechanical control mechanisms to achieve stable locomotion at high speeds (*Brown and Loeb, 2000*; *Jindrich and Full, 2002*; *Birn-Jeffery et al., 2014*; *Daley and Biewener, 2011*; *Daley et al., 2009*; *Frigon and Rossignol, 2006*; *Grillner, 2011*; *Lam and Pearson, 2002*; *More and Donelan, 2018*; *Pearson and Gramlich, 2010*; *Prochazka and*

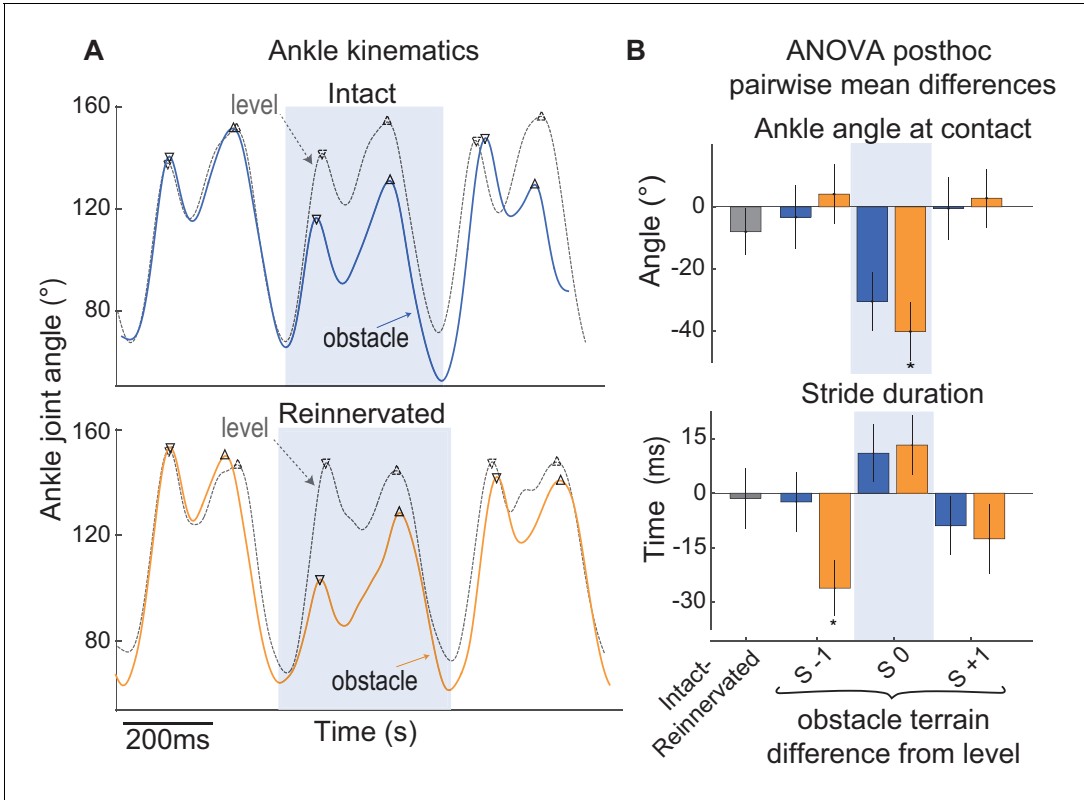

**Figure 8.** Ankle kinematics in guinea fowl with intact and reinnervated lateral gastrocnemius (LG). (A) Example ankle joint angle trajectories for a bird with intact LG (blue, top) and a bird with reinnervated LG (orange, below), running in obstacle terrain (solid lines) with level terrain (grey dashed lines). (B) Pairwise mean differences (mean ±95% ci) between intact and reinnervated treatment cohorts (grey), and obstacle stride categories compared to level terrain within each treatment cohort (intact: blue, reinnervated: orange). In obstacle strides (S 0, shaded box), the ankle is more flexed at foot contact in reinnervated compared to intact birds. Reinnervated birds show a shorter stride period in S −1, preceding the obstacle encounter, suggesting increased anticipatory preparation. (See *Figure 8—source data 1* for statistical results on ankle angle at the time of foot contact).

The online version of this article includes the following source data for figure 8:

**Source data 1.** ANOVA results for ankle angle at time of foot contact.

---

*Ellaway, 2012*). We hypothesized that an autogenic proprioceptive deficit will lead to increased reliance on feedforward tuning of muscle activity to achieve stable muscle dynamics in obstacle terrain. In birds with intact LG proprioception, the timing of muscle activity in obstacle-perturbed strides is consistent with combined feedforward and feedback control (*Daley and Biewener, 2011*; *Gordon et al., 2015*). Birds with reinnervated LG (rLG) exhibit a consistent phase shift in EMG onset relative to muscle length, with activation starting 6% earlier (23 ms) in the steady state contraction cycle, in both level and obstacle terrain (*Figure 6*, *Table 2*). This is consistent with a feedforward tuning of rLG activation timing to enable rapid force development and high muscle stiffness during stance, in the absence of monosynaptic reflexes. Regulation of EMG duration in obstacle strides (S 0) is absent in rLG (*Figure 5*), suggesting that proprioceptive feedback in late stance normally regulates force duration, which is disrupted following reinnervation.

A stable intrinsic mechanical response with neither feedforward- nor feedback-mediated changes in neural drive can occur when a perturbation is encountered at high running speeds (*Daley et al., 2009*). Rapid changes in muscle length and velocity in response to perturbations can decouple activation and force development (*Daley et al., 2009*; *Daley and Biewener, 2011*). In our previous work on intact in vivo muscle dynamics, variation in LG muscle strain during initial foot contact and limb loading explained 60% of the variation in force developed in obstacle encounters, while variation in LG muscle activation explained only 9%. This clearly demonstrates the decoupling between activation and force development that can occur in vivo (*Daley and Biewener, 2011*). These intrinsic mechanical effects minimize the disturbances in body dynamics that arise from terrain height

perturbations, enabling rapid recovery to steady gait. Similar intrinsic mechanical stabilizing responses have been demonstrated in the distal hindlimb joints of hopping and running humans subjected to unexpected changes in terrain height and stiffness (*Dick et al., 2019*; *Ferris et al., 1999*; *Moritz and Farley, 2004*). In concert with the stabilizing contributions of the intrinsic muscle-tendon dynamics, guinea fowl with intact proprioception also use feedforward and feedback regulation of muscle activity to maintain stability in obstacle terrain, with greater feedforward contributions when obstacles are visible and high contrast (*Daley and Biewener, 2011*; *Gordon et al., 2015*).

We find that guinea fowl with LG proprioceptive deficit achieve similar increases in total EMG activity during obstacle strides; however, the increases in activity occur early in the stride, before obstacle-induced changes in muscle force and length (*Figure 7*). This is consistent with anticipatory, feedforward increases in neural drive to the muscle, as observed in birds running over high-contrast visible obstacles (*Gordon et al., 2015*), and humans hopping on randomized but expected increases in surface stiffness (*Moritz and Farley, 2004*). These findings are consistent with a hybrid feedforward/feedback control model as conceptualized by *Kuo, 2002* in which feedforward and feedback gains are balanced to enable accurate state estimation and robust cyclical dynamics in the presence of both disturbances and sensory error. Although the reinnervated LG contributes to an effective obstacle negotiation response, it requires a longer recovery time and increased muscle activity following obstacle contact (*Figure 5B*). This suggests that the integrated response of the intact neuromuscular system enables robust stability with lower muscle activation costs.

Several features of the kinematics and muscle dynamics suggests coordinated plasticity and tuning of feedfoward control to compensate for reflex deficit following recovery from nerve injury (*Figure 9*). We observe similar increases in work output in response to obstacle encounters in rLG and iLG (*Figure 4B*). This finding is consistent with the idea that muscle work modulation is an important feature of task-level control for stability in uneven terrain (*Daley et al., 2009*; *Daley and Biewener, 2011*). However, work modulation is achieved through different underlying mechanisms in rLG and iLG. Earlier steady state activation in rLG (lacking autogenic proprioceptive feedback) enables higher muscle force development in early stance to resist the external load applied at foot contact, which likely contributes to the higher rate of shortening throughout stance. Additionally, rLG exhibits larger increases in peak force in obstacle encounters compared to iLG (*Figure 4B*). This increase in peak force likely involves both active and passive components: an active contribution from increased feedforward drive and EMG amplitude (*Figure 5B*), and a passive contribution from increased stretch of connective tissues associated with a more flexed ankle posture at foot contact (*Figure 8*). Finally, Birds with rLG also show an anticipatory shift in stride duration before obstacle encounters, which is not observed in the intact cohort (*Figure 8B*) and may help control landing conditions for obstacle encounters (*Gordon et al., 2015*). These findings suggest that reinnervated birds achieve effective muscle work modulation and stable obstacle negotiation through feedforward tuning of muscle activation and gait to compensate for loss of autogenic proprioception.

A recent study by *Sawicki et al., 2015* found that earlier onset of activation was associated with a shift to energy absorption in cyclical muscle contractions with a sinusoidal MTU length trajectory. We find here that earlier onset is associated with greater shortening and work production. The specific response of a muscle to a shift in activation phase is likely to be highly sensitive to the specific steady-state length trajectory of the muscle. Muscle force capacity and the activation and deactivation kinetics are substantially influenced by velocity and recent strain history, as demonstrated in controlled studies of in vitro muscle force-length work loops (*Askew and Marsh, 1998*; *Josephson, 1999*). Further work is needed to understand how in vivo muscle fascicle length dynamics interact with neural activation patterns and MTU compliance to enable tuning of muscle contraction dynamics to the mechanical demands of cyclical locomotor tasks.

We do observe shifts in muscle activity in late stance in some reinnervated individuals in response to obstacle encounters, which suggests heterogenic reflex responses in reinnervated LG (*Figure 2*). However, these responses are variable and idiosyncratic, with some birds showing increased EMG in late stance in obstacle strides, and others showing a decrease (*Figure 7—figure supplement 1*). The variable and idiosyncratic reflex responses result in wide confidence intervals for the obstacle-induced EMG response in the latter half of stance, despite consistent force-length trajectories over the same time-period (*Figure 7*). Idiosyncratic use of heterogenic reflex modulation across individuals following nerve injury recovery is consistent with findings in cats (*Lyle et al., 2016*; *Lyle and Nichols, 2018*). Guinea fowl have several agonist muscles to the LG that could contribute to

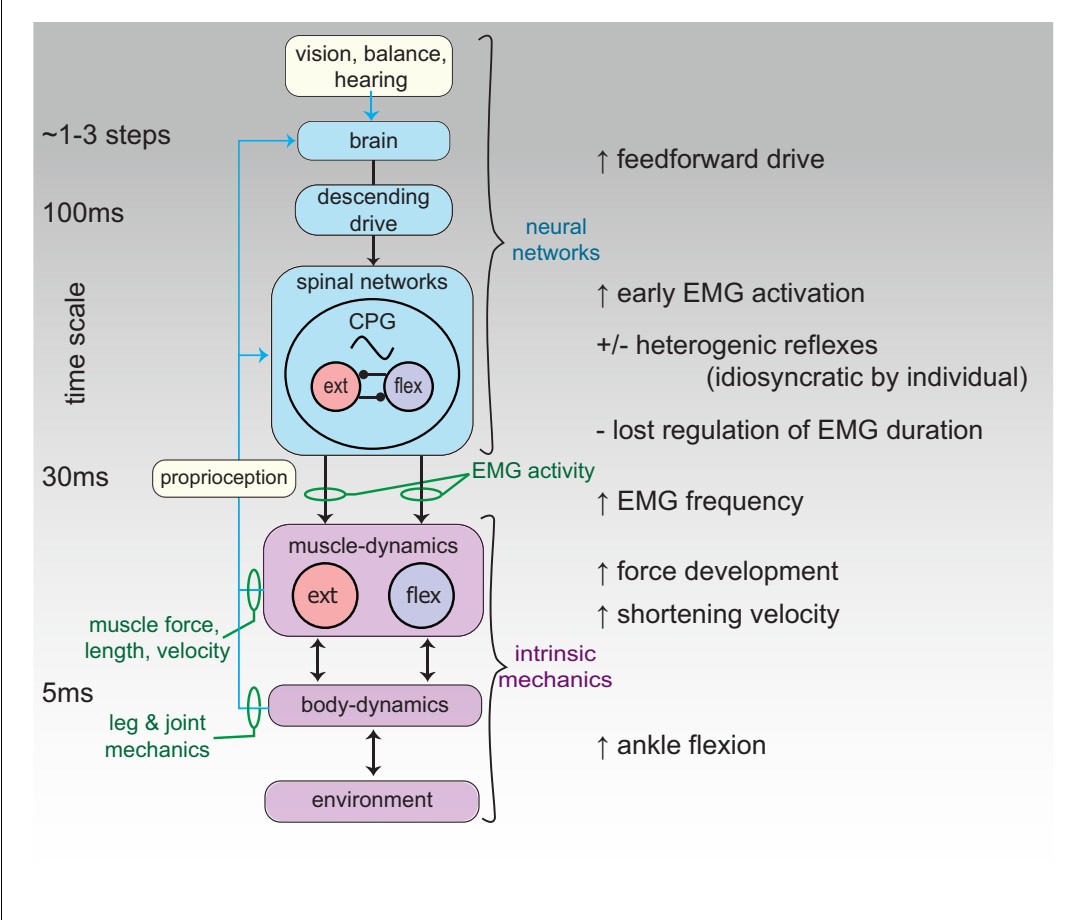

**Figure 9.** Schematic of the neuromechanical control mechanisms regulating function of the reinnervated lateral gastrocnemius (rLG) of the guinea fowl. Green text indicates the in vivo experimental measures used to infer sensorimotor control mechanisms. Differences in muscle dynamics between intact and reinnervated cohorts suggest that guinea fowl use a combination of feedforward and intrinsic mechanical mechanisms to compensate for disrupted proprioceptive reflexes, suggesting interconnected plasticity of neural and musculoskeletal mechanisms in the recovery from nerve injury.

heterogenic feedback modulation, including the medial gastrocnemius and digital flexors (*Daley and Biewener, 2011*; *Gordon et al., 2015*). However, no muscle is an exact synergist of the LG, because each has a unique combination of moment arms, fiber length, pennation angle and connective tissue compliance (*Daley and Biewener, 2003*; *Cox et al., 2019*). Consequently, it is unlikely that proprioception from agonists can completely restore accurate sensing to regulate LG force and work output. Additionally, in the presence of increased sensory error and noise, birds may learn over time to compensate through sensory integration in higher CNS pathways, leading to updated central coordination and feedforward drive to rhythm generating networks.

It was previously unknown whether guinea fowl would respond to reinnervation and proprioceptive deficit in a manner similar to quadrupedal mammals. We find that our results are consistent with the findings on rats and cats. Reinnervated cats and rats exhibit shifts in feedforward muscle activity and inter-joint coordination during slope walking, to compensate for loss of reflex-mediated ankle stiffness (*Abelew et al., 2000*; *Maas et al., 2007*; *Chang et al., 2009*; *Boeltz et al., 2013*). Cats and rats also preserve task level features of gait, such as leg length and body motions, despite variance in muscle and joint dynamics (*Chang et al., 2009*; *Boeltz et al., 2013*). These findings suggest performance of task-level goals as a target of sensorimotor optimization. Also similar to cats, guinea fowl exhibit variation among individuals in heterogenic compensation for loss of the autogenic stretch reflex, as suggested by the variable tendon tap responses and variation in late-stance EMG activity in obstacle perturbed steps (*Figure 7*; *Figure 7—figure supplement 1*). Work on cats suggests complex intermuscular feedback connectivity, which can recovery to varying degrees following

reinnervation (*Cope et al., 1994*; *Pearson, 2000*; *Lyle et al., 2016*; *Lyle and Nichols, 2018*). This complexity and variability among individuals following recovery from nerve injury reflects the complexity and plasticity of proprioceptive feedback networks. Nonetheless, the recovery of consistent task-level mechanical function supports the idea that sensorimotor control is optimized to maintain task-level performance goals such as stable body dynamics (*Chang et al., 2009*; *Safavynia and Ting, 2013*).

### Limitations and future directions

One of the major limitations in the current study is the potential for multiple differences between the intact and reinnervated cohorts that were not controlled, because the experiments on the two cohorts were conducted over different periods of time. The self-reinnervation procedure requires a long-term recovery period and results in a chronic sensory deficit, which is likely to lead to a complex array of changes in the musculoskeletal tissues and the sensorimotor networks. The current study did not include a sham-surgery experimental control, and we did not strictly monitor the ages of the original intact cohort at the time of the in vivo muscle-tendon surgeries. Nonetheless, it is reassuring that our findings are consistent with similar studies in cats and rats, suggesting feedforward tuning of muscle activation and ankle kinematics to maintain stability following loss of proprioception. In future experiments, it will be important to control the timing and amount of exercise training in intact and reinnervated experimental groups, considering the potential for exercise to influence the recovery process (*Boeltz et al., 2013*; *Brandt et al., 2015*).

The recovery process almost certainly involves coupled changes across multiple systems, including connective tissue compliance, muscle activation kinetics, fiber type distribution, motor unit size and distribution, spinal intraneuronal connectivity, and sensory integration in higher CNS centers for state estimation and movement planning. Due to the complex nature of these adaptations, it is challenging to fully tease apart individual contributions and mechanisms from in vivo experimental measures alone. In future studies, the coordinated mechanisms of sensorimotor adaptation and plasticity could be systematically explored through a combination of integrative experimental and computational approaches. These approaches could include 1) closed loop neuromechanical simulations to enable predictive hypothesis testing (*Ijspeert, 2014*; *Roth et al., 2014*), 2) combined use of in vivo measures of muscle dynamics with in vitro testing of muscle contractile dynamics, to replicate biologically realistic force-length contraction dynamics, 3) histological studies to examine changes in muscle fiber type distribution and connective tissue characteristics following reinnervation, and 4) perturbation approaches that probe both short and long term adaptation processes.

It remains unclear how the specific length trajectory and velocity features of in vivo muscle dynamics contribute to the intrinsic stability and control of movement. Dynamic measurement techniques are needed to address this challenge and to develop realistic models for in vivo muscle-tendon function. In addition to widely recognized force-length and force-velocity 'Hill-type' properties, muscle exhibits short and long-term history-dependent changes in force capacity in response to stretch and shortening (*Edman, 1975*; *Edman et al., 1978*; *Edman, 1980*; *Josephson, 1999*; *Herzog, 2004*; *Edman, 2012*; *Herzog, 2014*; *Rode et al., 2009*; *Nishikawa et al., 2012*; *Yeo et al., 2013*; *Nishikawa et al., 2018*). Recent developments in biorobotic platforms that enable controlled muscle experiments with realistic loading and length trajectories are promising tools for advancing our understanding of the role of intrinsic muscle dynamics in the control of movement (*Clemente and Richards, 2012*; *Richards, 2011*; *Robertson and Sawicki, 2015*). Integrative neuromechanical studies using multiple techniques will be essential for unravelling mechanisms of muscle function, sensorimotor integration and plasticity. Findings from these studies have important implications for many human health conditions, including acute nerve injury, diabetic neuropathy, neurodegenerative disorders, cerebral palsy, and muscular dystrophies.

## Materials and methods

### Animals and treadmill training

We obtained and reared six hatchling guinea fowl keets (*Numida meleagris*) from a breeder (Hidden Hollow Acres, Whitehouse Station, NJ), to allow re-innervation surgeries in juveniles with at least 12 weeks for recovery before in vivo muscle procedures (see below). At the time of the in vivo muscle

measurements, the guinea fowl had reached adult size, averaging 1.81±0.28 kg body mass (mean±S. D.). Birds had primary feathers clipped and were trained to run on a level motorized treadmill (Woodway, Waukesha, WI, USA). Training sessions were 15-20 minutes in duration, with breaks for 2 minutes as needed. All experiments were undertaken at the Concord Field Station of Harvard University, in Boston (MA, USA), and all procedures were licensed and approved by the Harvard Institutional Animal Care and Use Committee (AEP #20-09) in accordance with the guidelines of the National Institutes of Health and the regulations of the United States Department of Agriculture.

We also include data previously reported in *Daley and Biewener, 2011* from intact individuals (n=6, 1.77±0.63 kg body mass) to serve as a control group for statistical comparison to the new dataset (n=6 reinnervated individuals). We re-analyzed the intact cohort dataset alongside the reinnervated cohort, to ensure consistency in data processing and statistics. The intact data includes a larger sample than reported in *Daley and Biewener, 2011*, because the analysis here includes all strides in the level and obstacle terrain collected for running speeds between 1.3-2.0 ms$^{-1}$. Trials were recorded only for speeds that each bird could comfortably maintain on the treadmill belt for at least 30 seconds, allowing for 10-minute rest periods between trials with access to food and water. We focus on running speeds (>1.3ms$^{-1}$), to avoid the confounding effect of different sensorimotor control strategies in walking *versus* running (*Gordon et al., 2015*). Due to variation among individuals in the successful trials recorded, the intact cohort dataset includes a wider speed range (1.3-2.0 ms$^{-1}$) than the reinnervated cohort dataset (1.7-2.0 ms$^{-1}$). However, the analysis is focused on obstacle perturbations compared to steady gait at the same speed, and the datasets include comparable samples of obstacle encounters between the two cohorts: 128 for intact and 133 for reinnervated birds, respectively. In total, the dataset includes 1027 strides for reinnervated and 1512 strides for intact individuals and excludes 81 strides as outliers that were non-obstacle encounter strides with Z-score > 4. No obstacle encounter strides were excluded as outliers. The complete datasets for reinnervated and intact cohorts are available through DataDryad.org, including metadata and Matlab processing scripts (*Daley et al., 2020*, https://doi.org/10.7280/D11H49).

## Anesthesia and post-operative care

Birds were induced and maintained on a mid-plane of anesthesia using isoflurane (2 - 3%, mask/intubation delivery). We administered perioperative enrofloxacin and flunixin intramuscularly for analgesia after induction and continued for three days after each surgery. Birds recovered to bilateral weight bearing within 20 minutes following completion of surgical procedures.

## Reinnervation surgery

The timing of surgeries was planned based on a pilot study, which found full recovery of LG motor activity by 6 weeks following reinnervation surgeries, and continued absence of calcaneal tendon reflex one year later, indicating continued absence of autogenic stretch reflexes (*Carr et al., 2010*). We bilaterally transected and immediately repaired the peripheral nerve branch supplying the LG muscle in maturing guinea fowl between 7-12 weeks of age. We allowed time for full reinnervation recovery of motor output and growth to adult size before a subsequent surgery to implant muscle transducers (*Figure 1*). In the reinnervation surgery, a lateral incision was made posterior-distal to the knee to expose the underlying muscle. Blunt dissection enabled exposure and identification of relevant nerve branches, and the identity of the correct nerve branch was confirmed using an isolated nerve stimulator (SD48, Grass Instruments, Warwick, RI) to visualize contraction in the LG. After pre-placement of single longitudinal throw of 6-0 braided non-absorbable silk (Silk, Ethicon, Somerville, NJ, USA) through a 3 mm nerve section, we transected the nerve branch and sutured to appose the cut nerve endings. Fibrin glue (bovine thrombin in CaCl$_2$, fibrinogen, fibronectin from bovine plasma) was applied over the apposed nerve endings as an additional repair scaffold (*Carr et al., 2010*; *Spotnitz, 2010*). We closed the fascia and skin with 3-0 braided absorbable polyglactin (Vicryl, Ethicon,Somerville, NJ, USA).

In the immediate post-operative period, bilateral limb posture was visibly more crouched compared to 'intact' birds and Achilles tendon tap revealed no stretch reflex response. LG atrophy was qualitatively observed during the first 2 weeks of the recovery period. Within 1 week of surgery, bird activity levels appeared comparable to intact conspecifics, with limb posture partially recovered. From 2-3 weeks onwards, differentiating reinnervated from 'intact' birds was not possible from

grossly observable limb morphology, posture and gait. We conducted regular treadmill training from 7 weeks after reinnervation surgery. Exercise was started at 7 weeks to ensure synaptic withdrawal of primary afferents before recovery, eliciting a proprioceptive deficit in the self-reinnervated LG. In rats, synaptic withdrawal and resulting proprioceptive deficit is minimized if training is initiated on the 3$^{rd}$ day after nerve injury (*Boeltz et al., 2013*; *Brandt et al., 2015*). During training and experiments, birds did not stumble or fall with noticeably greater frequency than observed in intact birds and were able to maintain treadmill position over a similar speed range. At the time of muscle recordings, Achilles tendon tap revealed variable latencies of 49±43ms (mean±S.D., range 10.3-91.5ms). This may reflect variable recovery of intermuscular reflex connectivity, consistent with observations in cats and rats (*Boeltz et al., 2013*; *Brandt et al., 2015*; *Lyle et al., 2016*). In comparison, the Achilles tendon tap reflex latency in intact birds was 6.1±1.2ms (mean±S.D), consistent with the mono-synaptic stretch reflex (*Nishikawa et al., 2007*; *Daley and Biewener, 2011*).

## Transducer implantation surgery

When the birds were 23-28 weeks old (13-16 weeks following bilateral reinnervation surgeries), we performed a 2nd surgery for transducer placement, following similar procedures as *Daley and Biewener, 2003*. The surgical field was plucked of feathers and gently cleaned with antiseptic solution (Prepodyne, West Argo, Kansas City, MO, USA). We tunneled transducer leads subcutaneously from a 1–2 cm incision over the synsacrum to a second 4–5 cm incision over the lateral left shank. Sonomicrometry crystals (2.0 mm; Sonometrics Inc, London, Canada) were implanted into the lateral head of the gastrocnemius (LG) along the fascicle axis in the middle 1/3rd of the muscle belly. Crystals were placed in small openings using fine forceps, approximately 3–4 mm deep and 15 mm apart. We verified signal quality using an oscilloscope and secured the crystals by closing the overlying muscle fascia and lead wires with separate 4-0 silk sutures (Silk, Ethicon, Somerville, NJ, USA). Next to the crystal pair, we implanted bipolar EMG electrodes constructed from two strands of 38-gauge Teflon-coated stainless steel (AS 632, Cooner Wire Co., California, USA) with staggered 1 mm exposed regions spaced 1.5 mm apart. Electrodes were placed using sew-through methods and surface silicon anchors (3 x 3 x 2 mm) positioned with a single square knot at the muscle surface-electrode interface (*Deban and Carrier, 2002*). An "E"-type stainless-steel tendon buckle force transducer insulated with a polyurethane coating (Micro-Measurements, Raleigh NC) was implanted on the common gastrocnemius tendon, equipped with a metal foil strain gauge (type FLA-1, Tokyo Sokki Kenkyujo). We connected transducers to a micro-connector plug (15-way Micro-D, Farnell Ltd, Leeds, UK) sutured to the bird's dorsal synsacrum.

## Transducer recordings

A lightweight shielded cable was used to connect the microconnector to data acquisition systems. Sonomicrometry data were collected via a Sonometrics TRX analog data-acquisition device and PC interface (TRX Series 8, Sonometrics, Ontario, Canada). Crystals were tested before surgery in a saline bath to confirm distances measured by digital caliper matched those measured by the software. Occasional drop-outs and level-shift artifacts in the sonomicrometry length signal (arising from variation in signal-to-noise characteristics) were corrected within the Sonometrics software where possible and smoothed using cubic smoothing spline with a tolerance of 0.1 in MATLAB ('spaps' function, Mathworks, Inc; Natick, MA, USA). Tendon buckle signals were fed through a bridge amplifier (Vishay 2120, Micro-Measurements, Raleigh, NC), and EMG signals were amplified and bandpass filtered (10Hz and 3kHz) using GRASS pre-amplifiers (P511, Grass Instruments, Warwick, RI). Signals were recorded at 10kHz using a 16-channel, 16-bit Biopac A/D acquisition device (MP150, BIOPAC systems, Gotleta, CA, USA). Following experiments, birds were euthanized using an intravenous injection of sodium pentobarbital (100 mg kg$^{-1}$) while under deep isoflurane anesthesia (4%, mask delivery).

## Muscle morphology

Postmortem, we recorded the morphology of the muscle and the location of transducers to confirm muscle fascicle and tendon alignment. In the reinnervated cohort (this study) LG mass was 10.7±2.7 g and total gastrocnemius mass was 26.5±5.1 g. In the intact cohort (*Daley and Biewener, 2011*), LG mass was 10.2±4.3 g, and total gastrocnemius mass was 23.5±7.4 g. These muscle masses are a

comparable to those measured from intact guinea fowl in previous studies, representing approximately 0.5% body mass for LG and 1.3% body mass for total gastrocnemius mass (*Daley and Biewener, 2003*; *Higham and Biewener, 2008*). This suggests full recovery from denervation-induced muscle atrophy in the reinnervated cohort. Fascicle lengths for were 17 ±1 mm and 18±2 mm and pennation angles were 25±5° and 24±5° for reinnervated and intact LG, respectively. Crystal alignment relative to the fascicle axis ($\alpha$) was within 2°, indicating that errors due to misalignment were <0.1%. We calibrated the tendon force buckle in situ p*ost mortem* by applying a series of known cyclical loads using a force transducer (model 9203, Kistler, Amherst, MA), which yielded linear least-squares calibration slopes with $R^2 > 0.97$.

## Level and obstacle terrain conditions

We recorded trials on i) uniform level terrain and ii) terrain with repeating 5 cm obstacles, at the same treadmill speeds, as in *Daley and Biewener, 2011*. The treadmill belt (Woodway, Waukesha WI) was slatted black rubber-coated steel with running surface 55.8 cm x 172.7 cm with clearance for obstacles beneath. Obstacles were constructed from styrofoam reinforced with cardboard covered with black neoprene to form a light, stiff surface. Waterproof glue (Shoe Goo, Eugene, OR, USA) secured heavy-duty fabric hook and loop fastener (Velcro, Cheshire, UK) to the obstacle and treadmill surface. Four sequential slats of obstacles produced a 20 $cm^2$ continuous obstacle surface. Obstacles were encountered approximately every 4-5 strides, with some variation due to varied stride length and treadmill station keeping. We recorded high-speed video at 250 Hz (Photron, San Diego, CA, USA) for analysis of ankle kinematics, detection of stride timing and statistical coding of strides in relation to obstacle encounters.

## Data processing

We assigned strides categories in relation to the obstacle encounters, using the approach in *Daley and Biewener, 2011*, based on the stride sequence of the instrumented leg: the stride prior to an obstacle contact (S −1), obstacle contact strides (S 0), the stride following obstacle contact (S +1), and strides in flat terrain between obstacles (S +2). All level terrain strides were assigned the same stride category (L). Note, this is simpler stride coding than presented in *Gordon et al., 2015*, which also considered the timing of obstacle encounters by the contralateral leg. Strides in which the contralateral limb had made contact with the obstacle in the previous step are grouped here into the (S +2) category, for simplicity. This does not substantially alter the findings, because the coding was similar between intact and reinnervated birds, and the current analysis is focused on the shifts in LG force-length dynamics related to a direct obstacle encounter by the instrumented leg.

Features of LG activation, force-length dynamics and work output were measured, similar to *Daley and Biewener, 2011*. Raw EMG signals were used to calculate myoelectric intensity in time and frequency domains using wavelet decomposition (*Daley et al., 2009*; *Gordon et al., 2015*). This was used to calculate total myoelectric intensity per stride ($E_{tot}$) and mean frequency of muscle activation ($E_{freq}$). We calculated fractional fascicle length (L) from sonomicrometry data using mean length in level terrain as a reference length ($L_o$). Note, however, that $L_o$ is not directly related to sarcomere length or optimal length for the isometric force-length curve, which were not measured. Fractional fascicle length was differentiated to obtain fascicle velocity (V, in lengths per second, $Ls^{-1}$). Shortening strains are negative. We multiplied fascicle velocity (in $ms^{-1}$) by tendon force (in Newtons, N) to calculate muscle power (Watts), which was integrated through time to calculate total work per stride (Joules, J; with shortening work being positive), and then normalized by muscle mass to obtain mass-specific muscle work ($Jkg^{-1}$). We also recorded muscle (fascicle) length, velocity and force at specified times to evaluate how strain and activation factors influence muscle force and work output. All data processing was completed using MATLAB (Mathworks, Inc; Natick, MA, USA).

## Statistics

The statistical analysis approach is similar to that used in *Gordon et al., 2015* to investigate obstacle perturbation responses relative to steady state level terrain strides. We used a linear mixed-effects model ANOVA to test for significant effects of *treatment* (intact/reinnervated cohorts) and stride category (*stride ID*: Level, S -1, S 0, S +1, S +2) as fixed categorical factors, with individual (*ind*) included as a random effect. Statistical analysis was completed in Matlab using 'fitlme' and associated

functions in the Statistics and Machine Learning Toolbox (Mathworks, Inc; Natick, MA, USA). Several linear mixed effects models were evaluated:

1. 'Y ~ 1 + (1|ind)'
2. 'Y ~ 1 + treatment + (1|ind)'
3. 'Y ~ 1 + stride_ID + (1|ind)'
4. 'Y ~ 1 + stride_ID + treatment + (1|ind)'
5. 'Y ~ 1 + stride_ID *treatment + (1|ind)'

Model 5 (with the interaction term between fixed effects) was used as the final model, because it had the lowest AIC for 13 of 14 of the variables analyzed (AIC, Akaike, 1976). Model 4 had the lowest AIC for mean EMG frequency ($E_{freq}$), but the difference between Models 4 and 5 was not significant according to a likelihood ratio test. Therefore, for consistency, Model 5 was used for all variables. Posthoc pairwise comparisons were calculated for the mean difference ± 95% confidence interval between intact and reinnervated treatment cohorts and for the mean differences between level and obstacle strides categories within each treatment cohort. Pairwise comparisons were calculated after removing the random effect of individual, because the intact and reinnervated datasets came from different cohorts of individuals. We used False Discovery Rate to calculate an adjusted p-value threshold to maintain a 5% false positive rate across all statistical tests, including fixed effects tests and post-hoc pairwise comparisons (*Benjamini and Hochberg, 1995*).

## Acknowledgements

This research was supported by NIH grant NIAMS 5R01AR055648 to AAB, grant BB/H005838/1 to MAD from the Biotechnology and Biological Sciences Research Council (BBSRC), and a doctoral training studentship from the BBSRC to JCG supervised by MAD. Thanks to Jennifer A Carr for assistance in the experiments. Thanks to Reviewers N Cowan and L Ting and Reviewing Editor K VijayRaghavan for thoughtful and constructive feedback.

## Additional information

### Funding

| Funder | Grant reference number | Author |
|---|---|---|
| National Institutes of Health | NIAMS 5R01AR055648 | Andrew Biewener |
| Biotechnology and Biological Sciences Research Council | BB/H005838/1 | Monica A Daley |
| Biotechnology and Biological Sciences Research Council | Doctoral training studentship | Joanne C Gordon<br>Monica A Daley |

The funders had no role in study design, data collection and interpretation, or the decision to submit the work for publication.

### Author contributions

Joanne C Gordon, Formal analysis, Investigation, Methodology, Writing - original draft; Natalie C Holt, Investigation, Writing - review and editing; Andrew Biewener, Conceptualization, Resources, Supervision, Funding acquisition, Writing - review and editing; Monica A Daley, Conceptualization, Resources, Data curation, Software, Formal analysis, Supervision, Funding acquisition, Visualization, Methodology, Writing - original draft, Writing - review and editing

### Author ORCIDs

Andrew Biewener (ID) http://orcid.org/0000-0003-3303-8737
Monica A Daley (ID) https://orcid.org/0000-0001-8584-2052

### Ethics

Animal experimentation: All experiments were undertaken at the Concord Field Station of Harvard University, in Boston (MA, USA), and all procedures were licensed and approved by the Harvard

Institutional Animal Care and Use Committee (AEP #20-09) in accordance with the guidelines of the National Institutes of Health and the regulations of the United States Department of Agriculture. Surgery was performed under isoflurane anesthesia, and every effort was made to minimize suffering.

## Decision letter and Author response

Decision letter https://doi.org/10.7554/eLife.53908.sa1
Author response https://doi.org/10.7554/eLife.53908.sa2

## Additional files

### Supplementary files

• Transparent reporting form

### Data availability

The full dataset including raw data, metadata files and processing code have been deposited to Dryad (https://doi.org/10.7280/D11H49).

The following dataset was generated:

| Author(s) | Year | Dataset title | Dataset URL | Database and Identifier |
|---|---|---|---|---|
| Daley MA, Gordon JC, Holt NC, Biewener A | 2020 | Dataset for 'Tuning of feedforward control enables stable muscle force-length dynamics after loss of autogenic proprioceptive feedback' | http://dx.doi.org/10.7280/D11H49 | Dryad Digital Repository, 10.7280/D11H49 |

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
