## [Decision Letter]

**Acceptance summary:**

Gordon et al. use bilateral reinnervation of a leg muscle to deduce reflex contributions to muscle activity, kinematics, and muscle force-length trajectories during walking and obstacle crossing in bipedal birds, consistent with prior work in quadrupedal locomotion. Not only does the work advance our understanding, but the paper is very well-written with an extensive analysis that is of broad interest to those interested in complex sensorimotor integration. The work described in this paper is novel and its impact would be considerable in understanding the complexities of sensorimotor reflexes in bipedal locomotion.

**Decision letter after peer review:**

Thank you for submitting your article "Tuning of feedforward control enables stable muscle dynamics after loss of autogenic proprioceptive feedback" for consideration by *eLife*. Your article has been reviewed by two peer reviewers, and the evaluation has been overseen by K VijayRaghavan as the Senior Editor and Reviewing Editor. The following individuals involved in review of your submission have agreed to reveal their identity: Noah J Cowan (Reviewer #1); Lena H Ting (Reviewer #2).

The reviewers have discussed the reviews with one another and the Reviewing Editor has drafted this decision to help you prepare a revised submission.

Summary:

This paper studies the mechanism of sensorimotor integration of guinea fowl as an example of bipedal locomotion. Sensorimotor integration consists of neuronal feedforward, neuronal feedback and intrinsic muscle mechanics. Authors have studied how lateral gastrocnemius muscle (LG) and nervous system respond to the rapid perturbation when the proprioceptive sense as the fast feedback response is lost. Statistical comparison has been conducted between two groups (reinnervated and intact birds) through the recorded running trials at the same speed for both level and obstacle terrain. From the collected data of muscle force and length (intrinsic mechanics) and EMG (feedforward) signals, the authors have found that despite deficits in LG monosynaptic reflex following reinnervation, the general pattern of muscle dynamics of the reinnervated LG (rLG) is qualitatively similar to that of previously measured intact birds (iLG). However, the steady-state timing of EMG activation of rLG, as the feedforward, is phase-shifted earlier in the stride cycle. This earlier onset enables rapid force development and higher muscle stiffness at the time of foot contact and likely helps compensate for the loss of proprioceptive feedback. Additionally, regulation of force duration in obstacle strides is disrupted following reinnervation, suggesting that proprioceptive feedback in late stance normally regulates force duration. The authors have concluded that the autogenic proprioceptive deficit will lead to increased reliance on feedforward tuning of muscle activity to achieve stable muscle dynamics in obstacle terrain.

In general, this is a very well-written paper, with an extensive analysis that is of broad interest to those interested in complex sensorimotor integration. Also, in general, the figures and methods were very clear. The work described in this paper is novel and its impact would be considerable in understanding complexities of sensorimotor in bipedal locomotion.

There are some important concerns that, however, need to be addressed. In the main, these require:

A) Presenting the data more clearly.

B) A clear qualitative description of the differences between cohorts.

C) Redo the statistical analysis in a more interpretable manner.

D) Improve clarity of figures.

E) Reproducing data from the previous manuscript in order to make side-by-side comparisons will be convenient.

Essential revisions:

1) Suggestions:

Figure 2: To avoid confusion, it's better to use different colors since blue and orange are used extensively in this paper for iLG and rLG.

Figure 5: Figure is really busy and it's hard to visually see the differences between iLG and rLG. The difference between iLG and rLG looks negligible. It might be useful to show the mean and standard deviation of the difference between rLG and iLG.

2) A major weakness is the lack of discussion about potentially profound differences across the two groups, and inconsistent comparison across different cohorts of bird. While the authors do acknowledge that different birds were sampled in the intact and reinnervated groups in the Materials and methods, this point may easily be missed by the reader and should be re-stated as a limitation in the Discussion (perhaps starting at the seventh paragraph of the subsection “What is the role of proprioception in the control of high-speed locomotion?”). Considering that the nerves were injured while the birds were juveniles and testing was performed in adults means the normal maturation process may have impacted adaptation to proprioceptive loss and may limit applicability to understanding the roles of feedback and feedforward control, or effects of nerve injury, in adults. In general, similarities among groups are stated to be "comparable" without supporting quantification from the intact group (e.g. in the Materials and methods, body and muscle masses, muscle lengths, gait speed etc.; in Results, Figure 2. While is it also reported that tendon tap reflexes were absent, no quantification was provided. How are potentially different levels of recovery accounted for? Was exercise begun 7 weeks after injury to avoid regeneration associated with immediate training? (Brandt et al., 2015).

3) Despite the assertion in the Results (first paragraph), there are clearly differences in the level- and obstacle-terrain gait kinematics and kinetics between groups (c.f. Figures 2, 3, 6, 8).

4) Differences across S-1, S-0, and S+1 are interesting but not discussed. Why is it that S+1 has a greater advance in EMG than S-0 and S^-1^? This could be compared to downslope walking?

5) In general, figures were difficult to digest without careful parsing of the legends and could use better in-figure descriptors. For example, Figures 3 and 4 are not readable in black and white, the upper and lower rows could be labeled as "intact" and "reinnervated", while a dark line indicating "level terrain" with the colors indicating "obstacle". Foot contact lines could also be labeled, as the legends are hard to follow. Figure 3 legend "for a single individual from each condition" it is not clear whether "condition" refers to level vs. obstacle or intact vs. reinnervated.

6) The mixed model ANOVA results were difficult to follow, and it may be better to present specific individual findings based on the hypothesis, e.g. earlier LG activation, evidence of later reflex, There is also concern over the number of variables in the model and the potential for false positives (Tables 1, 2 and Figure 5). A select number of tests should be done in alignment with specific hypotheses posed. Mean effects should be reported using ANOVA and post hoc tests. The coefficients reported are not interpretable in terms of differences across conditions/steps (e.g. Figure 6B). Bar charts with the main effect would be easier to understand.

7) The implication for broader audiences could be enhanced, and the differences between bipedal and quadrupedal locomotor impairments are not clearly discussed. It is not clear that the shift to bipedal gait added significant insight into the role of proprioception in movement, as similar deficits occur in both.

---

## [Author Response]

In general, this is a very well-written paper, with an extensive analysis that is of broad interest to those interested in complex sensorimotor integration. Also, in general, the figures and methods were very clear. The work described in this paper is novel and its impact would be considerable in understanding complexities of sensorimotor in bipedal locomotion.There are some important concerns that, however, need to be addressed. In the main, these require:A) Presenting the data more clearly.

We have reformatted the figures to present data distributions and pairwise mean differences for the fixed effect categories. The figures, tables and text have been updated to allow direct comparison between the intact and reinnervated cohorts for all results.

B) A clear qualitative description of the differences between cohorts.

We have added text to the Materials and methods, Results and Discussion to provide clear descriptions of how the two cohorts compare in terms of experimental conditions (subsection “Animals and treadmill training”), reflex latency (subsection “Reinnervation surgery”), muscle morphology (subsection “Muscle morphology”), muscle function (subsections “Mechanical function of intact versus reinnervated LG”, “Force-length dynamics and work output during obstacle negotiation” and “Shifts in activation patterns between intact and reinnervated LG”) and gait kinematics (subsection “Stability and kinematic changes during obstacle negotiation in intact vs. reinnervated birds”). We revisit the differences between the cohorts in the limitations section of the Discussion (subsection “Limitations and future directions”). Throughout the text, we avoid stating that the two cohorts were ‘qualitatively similar’ or ‘comparable’ and instead state the specific similarities and differences between them. All figures provide direct comparison between the iLG and rLG cohorts, and we also provide the full datasets for both cohorts on DataDryad (https://doi.org/10.7280/D11H49).

C) Redo the statistical analysis in a more interpretable manner.

We revised the statistics as suggested to present pairwise mean differences between fixed effect categories in the figures and tables, with correction for multiple tests to maintain a 5% false discovery rate. We have also added a more detailed description of the statistical methods (subsection “Statistics”).

D) Improve clarity of figures.

We have modified the formatting of the figures as suggested by the reviewers, to consistently show the intact and reinnervated cohort datasets in direct comparison, with half-violin plots to show distributions, and bar plots with the mean and 95% confidence interval for pairwise comparison between fixed effect categories. We have added annotations to the figures to aid interpretation, as suggested by the reviewers.

E) Reproducing data from the previous manuscript in order to make side-by-side comparisons will be convenient.

All figures and analyses include the data from Daley and Biewener, 2011, alongside the new experimental data. We have provided the complete datasets (and processing code) for both cohorts through DataDryad as noted above.

Essential revisions:1) Suggestions:Figure 2: To avoid confusion, it's better to use different colors since blue and orange are used extensively in this paper for iLG and rLG.

Figure 2 has been revised to provide a direct comparison of average stride cycles for the intact and reinnervated birds. Annotations have been added to make the figure easier to read, and the color scheme is consistent with all other figures in the paper, with blue for intact gastrocnemius (iLG) and orange for reinnervated lateral gastrocnemius (rLG). (The current Figure 2 is a revised version of former Figure 4. The original Figure 2 has been updated and now provided as Figure 2—figure supplement 1).

Figure 5: Figure is really busy and it's hard to visually see the differences between iLG and rLG. The difference between iLG and rLG looks negligible. It might be useful to show the mean and standard deviation of the difference between rLG and iLG.

The original Figure 5 has been replaced with two separate figures (Figures 4 and 5) to allow clearer visual interpretation of the differences between iLG and rLG and to aid interpretation with respect to the hypotheses stated in the Introduction. Figure 4 contains half-violin distributions and bar plots for mean differences for variables relating to *muscle mechanical function*. Figure 5 contains half-violin distributions and bar plots for variables relating to *muscle activation*. Annotations have been added to aid the reader. The bar plots provide the mean and 95% CI for differences between the iLG and rLG treatment cohorts, and between level and obstacle terrain stride categories within treatment cohorts.

2) A major weakness is the lack of discussion about potentially profound differences across the two groups, and inconsistent comparison across different cohorts of bird. While the authors do acknowledge that different birds were sampled in the intact and reinnervated groups in the Materials and methods, this point may easily be missed by the reader and should be re-stated as a limitation in the Discussion (perhaps starting at the seventh paragraph of the subsection “What is the role of proprioception in the control of high-speed locomotion?”). Considering that the nerves were injured while the birds were juveniles and testing was performed in adults means the normal maturation process may have impacted adaptation to proprioceptive loss and may limit applicability to understanding the roles of feedback and feedforward control, or effects of nerve injury, in adults. In general, similarities among groups are stated to be "comparable" without supporting quantification from the intact group (e.g. in Materials and methods, body and muscle masses, muscle lengths, gait speed etc.; in Results, Figure 2. While is it also reported that tendon tap reflexes were absent, no quantification was provided. How are potentially different levels of recovery accounted for? Was exercise begun 7 weeks after injury to avoid regeneration associated with immediate training? (Brandt et al., 2015).

We have added text to the Materials and methods, Results and Discussion to provide clear descriptions of how the two cohorts compare in terms of experimental conditions (subsection “Animals and treadmill training”), reflex latency (subsection “Reinnervation surgery”), muscle morphology (subsection “Muscle morphology”), muscle function (subsections “Mechanical function of intact versus reinnervated LG”, “Force-length dynamics and work output during obstacle negotiation” and “Shifts in activation patterns between intact and reinnervated LG”) and gait kinematics (subsection “Stability and kinematic changes during obstacle negotiation in intact vs. reinnervated birds”). We revisit the differences between the cohorts in the limitations section of the Discussion (subsection “Limitations and future directions”. Throughout the text, we avoid stating that the two cohorts were ‘qualitatively similar’ or ‘comparable’ and instead state the specific similarities and differences between them.

We have also added additional text on the methods rationale and relevant citations to the literature on cats and rats.

All figures provide direct comparison between the iLG and rLG cohorts, and we also provide the full datasets for both cohorts on DataDryad (https://doi.org/10.7280/D11H49).

3) Despite the assertion in the Results (first paragraph), there are clearly differences in the level- and obstacle-terrain gait kinematics and kinetics between groups (c.f. Figures 2, 3, 6, 8).

The Results and Discussion have been updated to more clearly state the differences in gait kinematics between the intact and reinnervated cohorts (subsections “Stability and kinematic changes during obstacle negotiation in intact vs. reinnervated birds”, and “What is the role of proprioception in the control of high-speed locomotion?”).

4) Differences across S^-1^, S-0, and S+1 are interesting but not discussed. Why is it that S+1 has a greater advance in EMG than S-0 and S^-1^? This could be compared to downslope walking?

We summarize these differences between stride categories in the Results (subsection “Stability and kinematic changes during obstacle negotiation in intact vs. reinnervated birds”) and Discussion in the context of interpreting stability and neuromuscular control strategies (subsection “What is the role of proprioception in the control of high-speed locomotion?”). Although S +1 has slightly greater phase advance in activation, the difference is not statistically significant (Figure 6, Table 2). The main difference in S +1 for rLG is a 39% increase in total EMG activity, at 6% decrease in force duration and 4% decrease in stride duration, compared to steady state level strides (reported in the subsection “Stability and kinematic changes during obstacle negotiation in intact vs. reinnervated birds”).

We have added a paragraph to the Discussion to address how our findings relate to similar work on cats and rats (subsection “What is the role of proprioception in the control of high-speed locomotion?”).

5) In general, figures were difficult to digest without careful parsing of the legends and could use better in-figure descriptors. For example, Figures 3 and 4 are not readable in black and white, the upper and lower rows could be labeled as "intact" and "reinnervated", while a dark line indicating "level terrain" with the colors indicating "obstacle". Foot contact lines could also be labeled, as the legends are hard to follow. Figure 3 legend "for a single individual from each condition" it is not clear whether "condition" refers to level vs. obstacle or intact vs. reinnervated.

We have revised the figures as suggested, including added annotations to indicate ‘intact’ and ‘reinnervated’ cohorts, arrows indicating level and obstacle terrain strides, triangles for foot contact with an annotation arrow. We have revised the phrasing to refer to intact and reinnervated ‘*cohorts’* and level and obstacle terrain ‘*conditions’*.

6) The mixed model ANOVA results were difficult to follow, and it may be better to present specific individual findings based on the hypothesis, e.g. earlier LG activation, evidence of later reflex, There is also concern over the number of variables in the model and the potential for false positives (Tables 1, 2 and Figure 5). A select number of tests should be done in alignment with specific hypotheses posed. Mean effects should be reported using ANOVA and post hoc tests. The coefficients reported are not interpretable in terms of differences across conditions/steps (e.g. Figure 6B). Bar charts with the main effect would be easier to understand.

The figures have been revised to focus on specific variables relating to muscle mechanical function in Figure 4 and muscle activation in Figure 5, to make the link to the specific hypotheses clearer. (The hypotheses are stated in Introduction in the subsection “Investigating the role of proprioception through self-reinnervation”). The results have been re-sequenced to follow the revised figure sequence.

The statistics have been updated to provide post hoc pairwise mean differences between fixed effect categories from the ANOVA, with correction for multiple tests to maintain a 5% false positive rate. The figures include bar charts of the pairwise mean differences, with 95% confidence intervals. Comparisons are made between the intact and reinnervated cohorts (in grey) and between level and obstacle stride categories within treatment cohorts (in blue for iLG and orange for rLG).

7) The implication for broader audiences could be enhanced, and the differences between bipedal and quadrupedal locomotor impairments are not clearly discussed. It is not clear that the shift to bipedal gait added significant insight into the role of proprioception in movement, as similar deficits occur in both.

We have added a paragraph discussing the findings in the context of previous work in cats and rats (subsection “What is the role of proprioception in the control of high-speed locomotion?”). In general, our findings are consistent with the studies of quadrupeds. However, the studies of rats and cats have focused more on muscle activity patterns in relation to kinematics, without detailed analysis of in vivo muscle mechanical function (direct measures of muscle force and work). The current study provides insight into how shifts in muscle activation relate to shifts in muscle mechanical function to compensate for loss of reflex-mediated ankle stiffness.